# A Nonsense Variant in the *DMD* Gene Causes X-Linked Muscular Dystrophy in the Maine Coon Cat

**DOI:** 10.3390/ani12212928

**Published:** 2022-10-25

**Authors:** Evy Beckers, Ine Cornelis, Sofie F. M. Bhatti, Pascale Smets, G. Diane Shelton, Ling T. Guo, Luc Peelman, Bart J. G. Broeckx

**Affiliations:** 1Department of Veterinary and Biosciences, Faculty of Veterinary Medicine, Ghent University, B-9820 Merelbeke, Belgium; 2Small Animal Department, Faculty of Veterinary Medicine, Ghent University, B-9820 Merelbeke, Belgium; 3Department of Pathology, School of Medicine, University of California San Diego, La Jolla, CA 92093-0709, USA

**Keywords:** hypertrophic feline muscular dystrophy, Duchenne muscular dystrophy, disease-causing variant, nonsense

## Abstract

**Simple Summary:**

Muscular dystrophy (MD) in cats is a muscular disease that can have a fatal outcome. It is often caused by variants in the *DMD* gene and shows an X-linked recessive mode of inheritance. Therefore, females can carry such a variant without showing clinical signs of disease and can spread the variant to their offspring. These carriers can only be identified through DNA tests. Here, we identified the variant causing MD in two Maine coon cats. Maine coons can now be screened for this variant through a DNA test.

**Abstract:**

(1) Feline dystrophin-deficient muscular dystrophy (ddMD) is a fatal disease characterized by progressive weakness and degeneration of skeletal muscles and is caused by variants in the *DMD* gene. To date, only two feline causal variants have been identified. This study reports two cases of male Maine coon siblings that presented with muscular hypertrophy, growth retardation, weight loss, and vomiting. (2) Both cats were clinically examined and histopathology and immunofluorescent staining of the affected muscle was performed. *DMD* mRNA was sequenced to identify putative causal variants. (3) Both cats showed a significant increase in serum creatine kinase activity. Electromyography and histopathological examination of the muscle samples revealed abnormalities consistent with a dystrophic phenotype. Immunohistochemical testing revealed the absence of dystrophin, confirming the diagnosis of dystrophin-deficient muscular dystrophy. mRNA sequencing revealed a nonsense variant in exon 11 of the feline *DMD* gene, NC_058386.1 (XM_045050794.1): c.1180C > T (p.(Arg394*)), which results in the loss of the majority of the dystrophin protein. Perfect X-linked segregation of the variant was established in the pedigree. (4) ddMD was described for the first time in the Maine coon and the c.1180C>T variant was confirmed as the causal variant.

## 1. Introduction

Duchenne muscular dystrophy (*DMD*, MIM #310200) is an early-onset, rapidly progressive neuromuscular disease in humans. Primarily, male individuals are affected by *DMD* because of the X-linked recessive inheritance, and the prevalence ranges between 15.9 and 19.5 in 10,000 births [1]. It is a dystrophin-deficient muscular dystrophy (ddMD) caused by variants in the *DMD* gene. Symptoms are often attributed to the loss or significant shortening of the C-terminal domain, a region containing functionally important binding sites [2]. Generally, *DMD*-causing variants disrupt the translational ORF and lead to a premature termination codon, resulting in absent or non-functional dystrophin [3]. The most commonly reported variants are large deletions and duplications (66−72% and 9−14% of all cases, respectively), followed by nonsense variants (about 10% of all cases) [4,5]. Other types of variants that cause *DMD* are small indels, splice site, and missense variants (about 4%, 2%, and 1% of all cases, respectively) [4].

ddMDs, similar to *DMD*, exist in mice [6], rats [7], rabbits [8], pigs [9], dogs [10,11,12], cats [13,14], and monkeys [15]. The mdx mouse and golden retriever MD are commonly used as animal models for pre-clinical trials [16]. In dogs, ddMDs have been described in several dog breeds other than the golden retriever and, so far, 12 causal variants have been identified in this species. Comparable to humans, a wide variety of variant types and positions exists in dogs [17]. 

Because muscle hypertrophy is a prominent clinical sign in cats with ddMD, particularly of the tongue and diaphragm, the term hypertrophic feline MD has been used. However, selective muscle hypertrophy and pseudohypertrophy have also been described in humans with *DMD* [18] and in dogs with ddMD [19]. Therefore, the differentiation of a hypertrophic feline MD may not be appropriate in keeping with the one medicine approach. Similar to other animals, cats with ddMD display an X-linked recessive mode of inheritance, so feline X-linked MD (FXMD) would be a better term. So far, only two causal variants have been reported to cause FXMD in domestic shorthair cats, both large deletions involving one or more *DMD* promoters and one or more first exons [13,14]. Here, we describe the clinical and histopathological findings in two male Maine coon siblings with FXMD. Further molecular characterization led to the discovery of a new underlying causal variant.

## 2. Materials and Methods

### 2.1. Clinical Examination

Two Maine coon brothers were presented independently to the Small Animal Department of the Faculty of Veterinary Medicine of Ghent University with growth retardation, weight loss, hypersalivation, and vomiting. A general physical and complete neurological examination was performed on both cats. Complete blood counts and serum biochemistry including serum creatine kinase (sCK) activity were obtained in one cat and only sCK activity in the other cat. sCK activities were also obtained from the affected cats’ mother and two unaffected half-siblings. Electromyography and motor nerve conduction velocity studies were performed under general anesthesia in both affected cats. Electromyography recordings were taken from the facial, truncal, and appendicular muscles of the front and hind limbs. A commercially available electrophysiological unit (Natus Synergy UltraPro, Acertys Healthcare NV, Aartselaar, Belgium) was used for electrodiagnostic recordings. During the same anesthesia, echocardiography was performed in both cats and thoracic and abdominal radiographs were performed on one cat. Furthermore, skeletal muscle biopsies from the vastus lateralis and triceps muscles were obtained in both cats during the same anesthesia (standard biopsies as proposed by Dickinson and LeCouteur [20]). 

### 2.2. Sample Collection

Blood was collected in EDTA-laced tubes from both cases, their mother, an unaffected half-brother and -sister, and five unrelated unaffected Maine coon cats and stored at −20 °C. Biopsies from the vastus lateralis and triceps brachii muscles were obtained from both affected cats. About 1 cm^3^ of each sample was either prepared for histopathological and histochemical processing or gathered in a 1.5 mL test tube (Greiner Bio-One, Vilvoorde, Belgium), snap-frozen in liquid nitrogen, and stored at −80 °C for subsequent RNA-extraction (see below). A pedigree of the affected family with an overview of the available samples is depicted in Figure 1c.

### 2.3. Histopathology and Immunofluorescent Staining

To screen for morphological abnormalities in the affected cat muscle and verify the presence of dystrophin protein, unfixed chilled and formalin-fixed muscle biopsies from case 1 were shipped to the Comparative Neuromuscular Laboratory, School of Medicine, University of California San Diego. Subsequently, chilled muscle samples were flash-frozen in isopentane precooled in liquid nitrogen and processed by a standard panel of histochemical stains and reactions [21]. Formalin-fixed samples were routinely processed into paraffin. Cryosections were further processed by indirect immunofluorescence using monoclonal antibodies listed in Appendix A. Immunofluorescent staining of cryosections from the triceps muscle of case 1 was compared to an archived control muscle (limb muscle of a domestic short-haired cat of unknown age).

### 2.4. Muscle mRNA 

Total RNA was isolated from 50 mg muscle tissue (triceps brachii) of case 2 and an unrelated Maine coon cat without signs of muscular dystrophy using the AurumTM Total RNA Fatty and Fibrous Tissue Kit (Bio-Rad Laboratories, Hercules, CA, USA), according to the manufacturer’s spin protocol instructions. Subsequent cDNA synthesis was performed as described by Van Poucke et al. [22] and 2 µL was used as a PCR template.

#### 2.4.1. mRNA Expression

To inspect dystrophin mRNA expression in the affected muscle, five primer pairs were designed with NCBI primer-blast based on F.catus_Fca126_mat1.0, the latest feline reference assembly (Acc. No.: GCF_018350175.1; *DMD* location: NC_058386.1:g.26741260-29120869), taking into account secondary structures (mFold [23]), known SNPs (employ SNP handling in primer-blast), and repeat sequences (repeat filter set to automatic). For each primer pair, two identical PCR reactions were performed. The whole PCR product (10 µl) of one reaction was used to analyze the amplicons via gel electrophoresis, as described by Van Poucke et al. [22], whereas 2 µl of PCR product of the other reaction was loaded on gel electrophoresis, after which sequencing was performed with the remaining 8 µl, as described by Van Poucke et al. [22]. Primer sequences, amplicon length, and corresponding exon numbers for each amplicon are listed in Appendix A and details of PCR mixes and programs are presented in Appendix A.

#### 2.4.2. Variant Identification

Because the dystrophin muscle isoform is not annotated in F.catus_Fca126_mat1.0, the homology of the 20 described feline dystrophin protein isoforms was compared to the human dystrophin muscle isoform Dp427m (Acc. No.: NP_003997.2) using NCBI blast to identify the feline muscle isoform. Subsequently, the whole length of the respective transcript variant X8 (Acc. No.: XM_045050794.1) was sequenced, including the 5’- and 3’-UTR. Primer pairs were designed on the cDNA level, as described above. The 5’ and 3’ ends were sequenced using primers designed on gDNA. A 340 bp fragment in the 3’-UTR (XM_045050794.1:r.12596_12935) could not be sequenced because of the presence of long T-repeats. Primer sequences, amplicon length, PCR/sequencing mixes, and programs are listed in Appendix A.

### 2.5. Variant Confirmation and MYBPC3 Genotyping

To confirm the identified variant at the genomic level, DNA was isolated from EDTA blood, as previously described [24]. Five individuals from the family and five unrelated, unaffected Maine coon cats (controls) were genotyped for the variant through PCR with primer pair gDNA-3 (gDNA-F3 and gDNA-R3, Appendix A) and subsequent sequencing with gDNA-F3 (Appendix A). Variant segregation in the pedigree was inspected. The potential association between genotype and phenotype was determined by calculating the odds ratio (OR) with a 95% confidence interval (95% CI) and using a Fisher exact test. The effect of the variant on protein level was estimated with the PROVEAN online prediction tool [25] and MutPred-LOF web application [26].

Because the cardiac involvement was variable between both cases, they were genotyped for the A31P variant in *MYBPC3* that causes hypertrophic cardiomyopathy in the Maine coon [27]. The essay information for this genotyping can be found in Appendix A. 

### 2.6. Population Screening

The variant allele frequency in the Belgian Maine coon population was estimated by genotyping 95 EDTA blood samples (unrelated samples brought in by cat owners and veterinarians for routine genetic testing) from the biobank at the Animal Genetics Laboratory of the Faculty of Veterinary Medicine of Ghent University, as described above. Additionally, the variant allele frequency was determined in the general cat population based on the 99-Lives Consortium database [28] and the European Variation Archive (EVA) database [29]. 

### 2.7. Variant Classification

The standards and guidelines for the interpretation of sequence variants by the American College of Medical Genetics and Genomics (ACMG) were used to combine all of the aforementioned findings to classify the variant as (likely) pathogenic, (likely) benign, or of uncertain significance [30].

## 3. Results

### 3.1. Clinical Findings

Case 1 (Figure 1a,c: individual II1), a neutered male Maine coon, was presented at 5 months of age with growth retardation, weight loss, hypersalivation, and vomiting. No abnormal behavior was noticed by the owners. General physical examination revealed a mild systolic heart murmur grade 2/6 and prominent skeletal muscles (mainly gastrocnemius, caudal thigh, and triceps muscles) that appeared painful on palpation. An abnormal gait was present, consisting of a narrow stance with internal hock rotation in the pelvic limbs and a valgus stance in the thoracic limbs. Neurological examination was within normal limits. Spontaneous electrical activity consisting primarily of pseudomyotonic discharges with a typical “dive bomber” sound was present in all appendicular muscles (including masticatory and tongue muscles). Electroneurography and repetitive nerve stimulation were normal in both examined limbs. Echocardiography did not show any abnormalities besides a dynamic right ventricular outflow tract obstruction, causing an innocent heart murmur.

Case 2 (Figure 1b,c: individual II2), a male brother of case 1, was presented at 9 months of age with similar clinical signs. General physical and neurological examinations were within normal limits, apart from the same abnormal limb stance and prominent skeletal muscles described in case 1. Radiographs revealed cardiomegaly, marked hepatomegaly, and bilaterally increased volume of the caudal thigh muscles. Echocardiography revealed a trivial mitral and tricuspid insufficiency and no other abnormalities.

Complete blood count was normal for both cats. Biochemistry profiles showed a significant increase in sCK activities (13225 (case 1)–30850 U/l (case 2)), indicating a muscle disease. The unaffected mother (6 years old) and both unaffected half-siblings (4 years old) showed normal sCK activity (individuals I1, II3, and II4 in Figure 1b, respectively).

### 3.2. Histopathology and Immunofluorescent Staining

H&E stained cryosections of the triceps and vastus lateralis muscles (Appendix A) showed a dystrophic phenotype with variable myofiber sizes (diameters range: 13–114 µm), clusters of degenerating and regenerating fibers, and scattered calcific deposits that were highlighted by the alizarin stain (1), indicating a form of MD.

Immunofluorescence staining was used to determine a specific form of MD and is shown in Figure 2. Compared with control staining, sarcolemmal staining in the dystrophic muscle was not detected using antibodies against the rod (DYS1) and C-terminus (DYS2) of dystrophin. Staining for spectrin (SPEC2) and laminin α2 confirmed good tissue quality. Utrophin protein was increased in the dystrophic muscle’s sarcolemma (DRP2). Staining of α- and β-sarcoglycan was markedly reduced or absent. Groups of regenerating fibers were highlighted by the antibody against developmental myosin heavy chain (dMHC). These findings supported the diagnosis of ddMD.

### 3.3. Muscle mRNA Expression and Variant Identification

The presence of dystrophin mRNA was inspected by reverse transcription PCR, followed by the amplification of five dystrophin mRNA segments and gel electrophoresis. All five amplicons were found in the unaffected and affected muscle, confirming mRNA expression in the affected cats. The result for the affected muscle is shown in Appendix A and similar results were found for the unaffected muscle (not shown). 

Dystrophin isoform X8 was identified as the feline muscle isoform. Sequencing of the respective mRNA transcript variant X8 revealed eight variants. Of those variants, one is located in the 3’UTR (c.11088 + 408G > C) and six are synonymous variants (c.618A > G, c.2565G > A, c.2931G > A, c.6708A > G, c.8316T > C, and c.8949T > C). The remaining variant, NC_058386.1 (XM_045050794.1): c.1180C > T (p.(Arg394*)), is a nonsense variant in exon 11, leading to a truncated protein (Figure 3). This variant was predicted to be deleterious by PROVEAN, with a PROVEAN score of −5779.508; deleterious cutoff = −2.5) and pathogenic by MutPred-LOF, with a score of 0.45586 (pathogenicity threshold = 0.4 at a 10% false-positive rate).

### 3.4. Variant Confirmation, Population Screening, and MYBPC3 Genotyping

The c.1180C > T variant was confirmed at the gDNA level in both affected brothers. Pedigree analysis revealed perfect X-linked segregation after testing the healthy mother (heterozygous), half-brother (homozygous wildtype), and -sister (homozygous wildtype). The variant was absent in the five unrelated controls without signs of MD and diagnosed with disorders unrelated to MD. Because of perfect segregation, the OR was infinite. A significant association between genotype and phenotype (*p* = 0.02) was found using the Fisher exact test. The variant was not found in the 99-Lives Consortium database, EVA database, or Belgian Maine coon population.

Both cases were genotyped for the A31P variant in *MYBPC3* to discern if this could explain the difference in cardiac involvement between them. The variant was absent in both cats.

### 3.5. ACMG Classification

Altogether, six relevant criteria supporting classification as a pathogenic variant are fulfilled, whereas none of the criteria supporting a benign role are met (Table 1).

## 4. Discussion

This study reports two castrated male Maine coon littermates that were presented at a young age (5–9 months) with clinical signs of an early-onset inherited myopathy and markedly elevated sCK activities. Only a small number of congenital and dystrophic myopathies have been reported in cats including nemaline rod myopathy [31], X-linked myotubular myopathy in a Maine coon cat [32], laminin α2 deficient muscular dystrophy [33], muscular dystrophy associated with β-sarcoglycan deficiency [34], and X-linked ddMD [13,14]. Usually, sCK activities are normal or slightly elevated in congenital myopathies, making these disorders unlikely in the cats in this report. Immunofluorescent staining ruled out laminin α2 deficiency, but a β-sarcoglycan variant could not be ruled out. However, sarcoglycan variants result in autosomal recessive MDs and do not fit the X-linked pattern of inheritance seen here. As sarcolemmal staining was not detected using antibodies against dystrophin, the diagnosis of dystrophin-deficient FXMD was confirmed. 

Clinical signs in both FXMD-affected cats of this study were very similar to those in previously described domestic shorthair cases, with dystrophin deficiency including muscle hypertrophy as a prominent clinical feature and a markedly increased sCK activity as a clear indicator [13,14]. However, no hypertrophy of the diaphragm was noticed here, which is often seen in FXMD [35]. Case 1 showed progression of clinical signs, but was still alive at 3 years old at the time of this report. Case 2 died suddenly at home. Though this cat could have developed cardiomyopathy or fatal cardiac arrhythmia, clinical heart disease is rarely encountered in FXMD [35] and the underlying cause of death was not investigated in this cat. Of note, case 1 did display cardiac involvement and did not carry the *MYBPC3* A31P variant causing hypertrophic cardiomyopathy in this breed. While another (genetic) cause of hypertrophic cardiomyopathy cannot be excluded [36], this might be a case of FXMD with clinical heart disease. The variability in cardiac involvement makes the cat an interesting model to investigate MD modifier genes. 

In humans, female carriers of *DMD* variants are usually asymptomatic, but some do present with (mild) symptoms. About 8% of female carriers present with dilated cardiomyopathy [37]. Similarly, cardiac abnormalities, including enlargement and hyperechogenicity of the papillary musculature after the age of 2 years, were reported in obligate carrier females in a cat breeding colony [35]. Therefore, the owner of the cases’ mother was advised to present for regular echocardiographical monitoring. To date, no abnormalities have been found but, to be on the safe side, we advise monitoring other carriers of the c.1180C > T variant. Besides myocardial changes, many human female carriers also have increased sCK activity [38]. Similarly, a study in cats reported a mild increase in 2 out of 10 healthy female carriers, though the difference between the carrier and control group was not significant [39]. Here, the mother was the only healthy carrier available and her sCK activities were within normal limits. 

To date, only two disease-causing variants have been identified for FXMD, both large deletions in the *DMD* gene of domestic shorthair cats. The first deletion involves the muscle promotor and muscle and Purkinje neuronal first exon, whereas the second includes the cortical and skeletal neuronal promotors and their respective first exons. Both studies reported markedly reduced or absent levels of dystrophin in affected skeletal muscle, as was the case in the current study [13,14]. Contrary to reports of Winand et al. [14], the whole length of the muscle transcript was present in the dystrophic cat muscle of case 2.

The mRNA expression in the affected muscle is not surprising, as this has been reported before in *DMD* patients and mdx mice with premature termination codons, albeit greatly reduced [40,41]. Here, no (semi)quantitative analysis was performed on dystrophin mRNA expression because of sample size limitations. Of note, visual inspection of the amplicon fragments after loading 10 µl of PCR product from the affected muscle onto a gel (Appendix A) revealed a comparable intensity to that of unaffected muscle, suggesting that mRNA expression is not extremely reduced in the FXMD cats from this study. 

To confirm the differential diagnosis of FXMD and rule out any other congenital myopathies, a histopathological evaluation of muscle biopsies was performed. Immunohistochemical staining showed absent protein localization for the dystrophin rod-domain and C-terminus in dystrophic muscle. Because commercially available antibodies for humans were used and the antibody against the N-terminus is human-specific, no staining was performed against this region. The presence of the N-terminal domain cannot be excluded as the c.1180C > T variant is positioned just downstream of this region. A small amount of dystrophin has already been reported in the affected muscle of a cavalier king Charles spaniel with a 7 bp deletion, resulting in a premature termination codon in exon 42 [42]. Likewise, multiple human dystrophinopathy patients showed normal or reduced immunohistochemical staining for one or more dystrophin domains, though staining for the C-terminus is typically negative in *DMD* [43].

Staining for α-, β-, and γ-sarcoglycans was markedly reduced or absent in dystrophic muscle from case 1. The sarcoglycan subcomplex is part of the dystrophin-associated protein complex, allowing dystrophin to interact with the sarcolemma, cytoskeleton, channel proteins, and signaling or scaffolding proteins [44]. The reduced sarcoglycan staining is likely secondary to dystrophin deficiency [43]. Likewise, an increased utrophin staining of the FXMD muscle is also secondary to dystrophy and is often seen in *DMD* [43,44]. Utrophin is homologous to dystrophin and can recruit many proteins involved in the dystrophin-associated protein complex. Normally, the protein’s expression is downregulated by dystrophin but, as dystrophin is downregulated in ddMD individuals, utrophin expression is upregulated [44].

The absence of dystrophin protein by immunofluorescence staining in FXMD muscle made *DMD* an obvious candidate gene for genetic analysis. Because of the expense of whole-genome sequencing, the enormous size of the *DMD* gene, and the availability of mRNA in the affected muscle samples, variant identification was performed at the mRNA level. After performing RT-PCR and sequencing the entire dystrophin cDNA, the most likely disease-causing variant was identified as NC_058386.1 (XM_045050794.1): c.1180C > T (p.(Arg394*)), a nonsense variant in exon 11 leading to a truncated protein. Combining the aforementioned results, we evaluated whether the newly identified variant could be classified as pathogenic based on the ACMG guidelines. The first important criterion (PVS1) is a nonsense variant where loss-of-function is a known mechanism of disease [30]. Nonsense variants are known to cause 10% of *DMD* cases in humans [4,5] and the causal variant in the mdx mouse is also a nonsense variant [45]. A recent study looked at the impact of the position of nonsense variants in the *DMD* gene on the phenotype. The authors found that nonsense variants in 51 exons, including exon 11 (which corresponds to exon 11 in feline muscle transcript variant X8), were associated with *DMD* 100% of the time [46]. In evaluating the evidence level of a nonsense variant, the presence of multiple gene transcripts must be considered. The latest feline reference assembly predicts 20 transcript variants. The c.1180C > T variant would not affect transcript variants X12–X20, which are predicted to create shorter protein isoforms. These shorter isoforms are comparable to human Dp260, Dp140, Dp116, and Dp71, none of which play important roles in muscle tissue [47]. Therefore, they are not biologically relevant to the disease phenotype. Altogether, the nonsense nature of the c.1180C > T variant is very strong evidence (PVS1) of pathogenicity [30].

Strong evidence of this variant’s pathogenicity is provided by the abundance of functional studies (PS3) on the human *DMD* gene [44]. Additional strong evidence is delivered by the variant’s increased prevalence in affected cats over controls (PS4), measured by the infinite OR. As a CI cannot be calculated with an infinite OR (because of perfect segregation), a *p* value (*p* = 0.02) was calculated using a Fisher exact test to demonstrate a significant association between genotype and phenotype. The absence of the variant in multiple population cohorts yields further moderate evidence (PM2). While the cases’ mothers’ importation from Italy might explain the absence of the variant in the Belgian Maine coon population, the variant was also absent from the 99-Lives Consortium database, which contains samples from all over the world. Whether the variant is absent from the Italian Maine coon population could not be determined in this study. Finally, supporting evidence is provided by perfect segregation of the variant with the FXMD phenotype demonstrated by the pedigree analysis (PP1) as well as two in silico predictive programs estimating a deleterious effect (PP3). When one very strong (PVS1) and one or more strong (PS1–PS4) criteria are fulfilled, a variant can be classified as pathogenic, as proposed by the ACMG [30]. Not only did we provide very strong (PVS1) and strong evidence (PS3 and PS4), but additional moderate (PM2) and supporting evidence (PP1 and PP3) is also available. Furthermore, none of the criteria supporting a benign role are met. Consequently, we can conclude there is abundant evidence that the c.1180C > T variant is pathogenic.

## 5. Conclusions

Overall, this study is the first to report ddMD in the Maine coon. It describes the dystrophic phenotype, absence of dystrophin by immunofluorescence, and the disease-causing variant, with ample evidence of pathogenicity and the effect at the transcript and protein level. Furthermore, X-linked segregation was confirmed based on pedigree analysis and the variant was absent in three databases.

## Figures and Tables

**Figure 1 animals-12-02928-f001:**
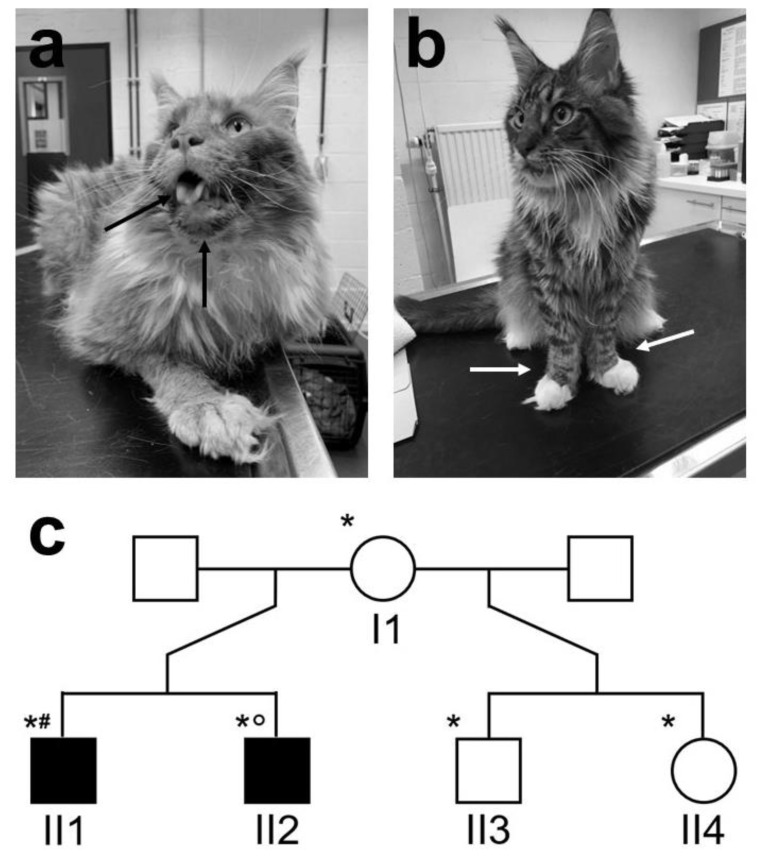
Image of affected cases with pedigree. (**a**) Case 1 (individual II1 in c) at the age of 3 years old, showing tongue hypertrophy, visible in the mouth and beneath the mandibula (black arrows). (**b**) Case 2 (Individual II2 in c), showing internally rotated carpi of the thoracic limbs (white arrows). (**c**) Pedigree with numbering of the individuals. Sample availability is indicated by a * for an EDTA blood sample, a ° for a muscle biopsy for genetic analysis, and a # for a muscle sample for histopathology and immunofluorescent staining.

**Figure 2 animals-12-02928-f002:**
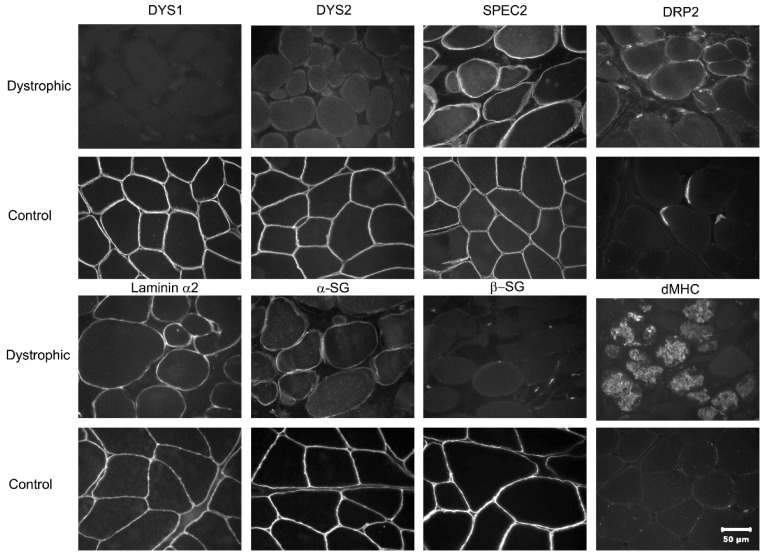
Immunofluorescent staining of dystrophic and control muscle. Immunofluorescent staining of muscle cryosections showing localization of dystrophin and laminin α2. Staining was performed using monoclonal antibodies against the rod-domain (DYS1) and C-terminus (DYS2) of dystrophin, spectrin (SPEC2), utrophin (DRP2), laminin α2, α- and β-sarcoglycan (α- and β-SG, respectively), and developmental myosin heavy chain (dMHC). The magnification bar in the lower right = 50 µm for all images.

**Figure 3 animals-12-02928-f003:**
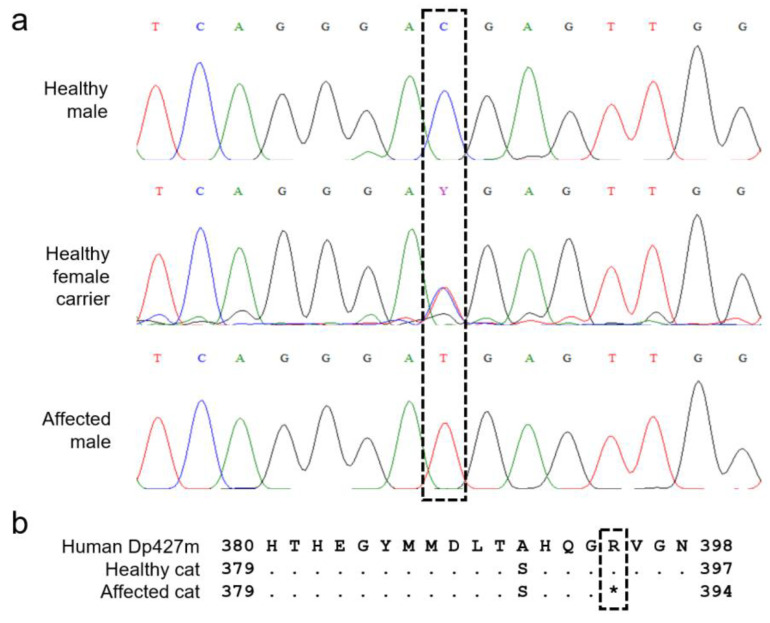
Disease-associated variant. (**a**) Chromatogram alignment of a healthy male, a healthy female carrier, and an affected male. (**b**) Protein alignment of the sequence around the variant. A * indicates the end of the protein.

**Table 1 animals-12-02928-t001:** Relevant criteria for variant classification.

Criterion	Result	Remarks	Conclusion
Null variant in a gene where LOF is a known mechanism of disease (PVS1)	LOF confirmed at protein level	Presence of alternate gene transcripts, but the variant affects the biologically relevant transcript	PVS1 fulfilled
Functional studies show deleterious (PS3) or no deleterious (BS3) effect	Abundance of functional studies of the human *DMD* gene show a deleterious effect		PS3 fulfilled
Significant OR > 5 (PS4)	OR = infinite; *p* = 0.02	CI could not be calculated because of perfect segregation, so *p* value was calculated	PS4 fulfilled
Variant absent in population databases (PM2)	Variant absent in two databases and a Belgian Maine coon cohort		PM2 fulfilled
Co-segregation in multiple affected family members (PP1)	Perfect variant segregation with FXMD phenotype in the pedigree	Limited sample availability, but segregation confirmed in two affected members	PP1 fulfilled
Multiple lines of computational evidence (PP3)	2/2 programs predicted deleterious/pathogenic effects		PP3 fulfilled

## Data Availability

The data generated during this study can be found within the published article and its Appendix A. The variant data for this study have been deposited in the European Variation Archive (EVA) at EMBL-EBI under accession number PRJEB52511. Additional data are available from the corresponding author upon reasonable request.

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
