# Peer review of "A Nonsense Variant in the DMD Gene Causes X-Linked Muscular Dystrophy in the Maine Coon Cat"

_animals, 2022, doi:10.3390/ani12212928_

Round 1

Reviewer 1 Report

This article describes the identification of a variant in exon 11 of the gene coding for dystrophin in cats, which induces a nonsense mutation causing a premature stop in the translation of the protein. Indeed, the protein is almost absent on the histological sections made from muscle biopsies of one of the two affected cats.
The destabilization of the dystrophin-associated complex proven by immunostaining, the segregation of the variant consistent with the phenotype of animals of the proband nuclear family and the absence of the variant in the sequences of control cats available in public databases all support the hypothesis proposed by the authors of a causative variant affecting the two clinically characterized cats.
The evidence provided is convincing and supports the authors' conclusion which I share.

The article is concisely written and pleasant to read.

To be published, I would recommend some significant changes and improvements.

1- First, the updating of a section of the bibliography. For example, reference [8] is cited - the first time line 51 - as a journal listing canine models of DMD. Since 2012, at least 7 papers have been published on newly identified variants. Therefore, it is important to cite a more recent review on the subject or to add the missing references. The same remark applies to reference [6] for mice, published in 1999. Today, there are at least 14 mouse models, of which 8 have been published in the last 5 years. In the context of this introduction, which intends to list all mammalian models for DMD, it is also important not to ignore the rat model - at least 4 models published since 2014 - as well as the rabbit and monkey models.

2- I also recommend including the laminin labeling images in Figure 2. It is important for the reader to have visual evidence of the integrity of a muscle that has traveled before being frozen. Spectrin labeling partially meets this expectation, but laminin labeling would likely provide an additional element of conviction.

3- A Western blot image, complementary to in situ immunodetection, would also be useful.

4- Finally, it would be very interesting to specify the genotype of the two affected cats at the MyBPC3 locus. It seems that the cardiac involvement is very variable between the two cats. Could this interindividual variability result from the interference of an A31P mutation in one of the two and, depending on the outcome, contribute to a worsening or, on the contrary, a transient protection against the heart disease induced by the DMD mutation? This data is worth describing and interesting to discuss. In case of identical genetic status between the two cats for MyBPC3, it would then be interesting to discuss the interest of outbred models such as the cat for the identification of disease modifier genes.

Minor points & correction of typos

The DMD gene should be italicized everywhere (e.g. lines 43, 314, 336).

Line 46, I suggest: resulting in absent or non-functional protein

Lines 83: please explain why these 2 muscles were chosen

Line 110: please include breed and age of controls, as well as which muscle was used

Paragraphs 2.2 and 2.4.1: add a space between numbers and units. Also line 287

Line 164-167 to be removed

Line 169: Figure 1b should be 1c

Line 175: include behavioural data from the owner, if available

Line 181: Figure 1b should be 1c

Lines 209 and 210: to be removed, pictures included in Figure 2

Line 217: Define what the 5 amplicons are. This info needs to be clarified for readers who will not read the M&M section

Line 232: indicate the genotype of the half-brother between brackets, as for the 2 other cats

Line 286: “Of note” rater than “However”, which tends to make the visual inspection a proof.

Line 288: “Suggesting that” rather than “indicating”

Line 315: vVariant replaced by variant

Line 339: remove one space after p-value (I think there are two)

Reviewer 2 Report

This is an excellent contribution to our understanding of MD in cats. The diagnosis is clearly established, the study setup including unaffected relatives and controls is sound, the molecular methods are appropriate and the results convincing. I have no suggestions on improving this manuscript.

Author Response

Dear reviewer,

Thank you for the time you put into thoroughly reading and reviewing our manuscript. Some changes were made after suggestions from the other reviewers. They are indicated with track changes. The biggest changes are listed below:

* The bibliography was updated in the introduction.

* Figure 2 was changed: Laminin staining is now shown instead of γ-sarcoglycan.

  • Both cases were genotyped for the A31P variant in MYBPC3, causing hypertrophic cardiomyopathy. The cats were both homozygous wildtype. Details of the genotyping essay were added to Table S6.

Best wishes,

Evy Beckers

Reviewer 3 Report

The manuscript submitted by Beckers et al. entitled "A nonsense variant in the DMD gene causes X-linked muscular dystrophy in the Maine coon cat" identify two cases of Feline dystrophin-deficient muscular dystrophy for the first time in the Maine coon cat and the c.1180C>T variant was confirmed as the causal variant.

The manuscript is well-written and the results are nicely presented and discussed. The experiments are also well-design.

The authors need to remove the text between lines 165-167 and, after this, the paper is in a suitable form to be published in the opinion of this reviewer.

Author Response

Dear reviewer,

Thank you for the time you put into thoroughly reading and reviewing our manuscript. We deleted the lines in the beginning of the results. Some additional changes were made after suggestions from the other reviewers. They are indicated with track changes. The biggest changes are listed below:

  • The bibliography was updated in the introduction.
  • Figure 2 was changed: Laminin staining is now shown instead of γ-sarcoglycan.
  • Both cases were genotyped for the A31P variant in MYBPC3, causing hypertrophic cardiomyopathy. The cats were both homozygous wildtype. Details of the genotyping essay were added to Table S6.

Best wishes,

Evy Beckers